# DICES Dataset:
# Diversity in Conversational AI Evaluation for Safety

Lora Aroyo[1], Alex S. Taylor[2], Mark Díaz[1], Christopher M. Homan[1], Alicia Parrish[1], Greg Serapio-García[3], Vinodkumar Prabhakaran[1], and Ding Wang[1]

[1]Google Research, [2]City, University of London, [3]University of Cambridge

## Abstract

Machine learning approaches often require training and evaluation datasets with a clear separation between positive and negative examples. This risks simplifying and even obscuring the inherent subjectivity present in many tasks. Preserving such variance in content and diversity in datasets is often expensive and laborious. This is especially troubling when building *safety* datasets for conversational AI systems, as safety is both socially and culturally situated. To demonstrate this crucial aspect of conversational AI safety, and to facilitate in-depth model performance analyses, we introduce the DICES (Diversity In Conversational AI Evaluation for Safety) dataset that contains fine-grained demographic information about raters, high replication of ratings per item to ensure statistical power for analyses, and encodes rater votes as distributions across different demographics to allow for in-depth explorations of different aggregation strategies. In short, the DICES dataset enables the observation and measurement of variance, ambiguity, and diversity in the context of conversational AI safety. We also illustrate how the dataset offers a basis for establishing metrics to show how raters' ratings can intersects with demographic categories such as racial/ethnic groups, age groups, and genders. The goal of DICES is to be used as a shared resource and benchmark that respects diverse perspectives during safety evaluation of conversational AI systems.

## 1 Introduction

As conversational AI systems built on large language models (LLMs) have become increasing prevalent, it is clear that their safety should be of paramount importance [29, 39]. Safety, in this context, speaks to the character of automatically generated content and its capacity to cause dangers to downstream users/communities, propagate mis/dis-information, and violate social norms in respectful communication. While advances in conversational AI capabilities are being propelled by access to web-scale, unlabelled training data [10] and innovative modelling approaches [40], such forms of safety continue to demand robust evaluation data and careful fine-tuning to align with social norms [39] and achieve the responsible practices rightly demanded across the tech sector [31, 14]. In other words, carefully curated data remains crucial to ensuring safe deployment of conversational AI [45].

Recent research has investigated approaches to efficiently fine tune language model outputs using safety annotated datasets [37, 39, 24, 15, 19, 28, 46]. Much less work, however, has gone into understanding the inevitable complexities associated with building datasets that capture the range of views of safety across diverse populations. Existing resources have typically disregarded the varied and subjective ideas of safety in conversation held by diverse user groups; instead, they adopt a simplified notion of a single ground truth for safety in order to facilitate technical solutions – an approach that can have unwanted or even disastrous effects in real-world settings [5, 21]. Specifically, much of the work addressing safety has neglected the variation between different populations that

may hold diverse perspectives on safety [6?, 7], and there is little in the way of guidance for building or evaluating resources that capture such diversity.

In this paper, we introduce a new dataset, DICES (*Diversity In Conversational AI Evaluation for Safety*), that offers a framework for the fine-grained representation and analysis of safety perceptions from different user populations.[1] DICES incorporates safety ratings of human-bot conversations by a large, demographically diverse set of human raters. The dataset has also been designed from the ground up to capture balanced proportions of opinions from sub-populations in the rater pool (for gender, age and ethnic/racial groups) and to generate an exceptionally large number of ratings per conversation. This provides a particularly powerful means for evaluating safety in language models, especially with respect to population diversity.

## 1.1  Contributions

Set against the urgent need for more nuanced approaches to safety in language modelling and, in particular, a more systematic way to account for diverse opinions of safety in model training and evaluation, the primary contributions of this paper and the DICES dataset are as follows:

**Rater Diversity** – Intentionally diverging from the language of bias (and attempts to mitigate it), we approach differences between raters opinions of safety in terms of diversity (for a brief summary of related work on rater diversity, see [43]). This is important because it motivates our aim to characterise the impact of raters' backgrounds on dataset annotations produced and used by large language models. We avoid assuming bias should, by default, be mitigated or removed, as it is likely the case that differences within a population will be crucial to the real-world efficacy of models. DICES is thus intentionally designed to account for diversity, with a rater pool that has a balanced distribution across demographic groups.

**Expanded Safety** – In line with the arguments in favour of dialogue safety ratings [e.g., 38] and extending the approach in Thoppilan et al [39], we assess a wider notion of safety that includes detailed ratings along five safety categories related to harm, bias, misinformation, politics and safety policy violations. The DICES dataset offers a means of evaluating the safety of conversational AI systems as well as focusing on granular categories such as specific harms, biases, dangerous content, hate speech and misinformation (and how they intersect with different demographic groups).

**Dataset Size** – DICES contains two sets of annotated AI chatbot conversations – one of 990 conversations (DICES-990) and one of 350 (DICES-350) with approximately 70 and 120 annotations per conversation, respectively. This rater replication rate is exceptionally large, as most annotation tasks use only 3–5 raters per conversation. The high number of unique ratings per conversation allows for statistical power of the observations drawn from the data. This is especially important for the purposes of adequately studying the demographic diversity of annotators and any impact on their safety opinions. In addition, *the high number of unique ratings per conversation also allows for resampling of the data to model results at more typical sample sizes with a better estimation of variability*. Both datasets include multiple sub-ratings which specify the type of safety concern, such as type of hate speech and the type of bias or misinformation, for each conversation. Both datasets have been developed using a systematic approach to recruiting raters from different demographics. DICES-990 was annotated by raters recruited from two locales (India and US) and distributed across genders. DICES-350 was annotated by raters only from the US and distributed in a balanced way across gender, ethnicity and age group. We concentrate on DICES-350 due to space constraints, however both datasets are released with this paper.

**Metrics** – Next to presenting DICES, we also illustrate how it can be used to develop metrics to examine and evaluate conversational AI systems in terms of both safety and diversity. For example, we show how measures of inter-rater reliability can be compared to reveal different agreement results between demographic subgroups (see Figure 4). Our dataset repository includes a more complete set of metrics and results from their application.

## 2  Related Work

In dataset creation for the evaluation of language models, the detection of toxicity, harm and hate speech is receiving increased attention, particularly in light of the recent popularity of LLMs (see related surveys by [4], [12] and [16]). Broadly, this line of research aims to assess the degree of toxicity, harm or hate speech in datasets or to evaluate a machine learning model's propensity to either identify or reproduce such language [e.g., 9, 20, 34, 36, 37].

---

[1]DICES dataset and accompanying analyses: https://github.com/google-research-datasets/dices-dataset/

Crucial to much of this work is the role of labelled data, *hand-labelled by human raters*. Human annotation continues to contribute to a growing number of datasets used for benchmarking toxicity, harm or hate speech and to train/fine-tune language models [e.g., 27, 44]. However, an acknowledged risk of these datasets and their corresponding models is that they have the potential to exhibit bias [5, 26, 32, 42], in part because of raters' backgrounds and experiences and how these, in turn, impact opinions [? 13, 35]. For instance, raters' demographic markers (i.e., their first language, age, and education) have been shown to have a significant impact on the performance of models that detect hate speech and abusive language [3]. Similarly, with a narrower focus of African American and LGBTQ populations, Goyal et al. [17] have shown statistically significant differences between toxicity ratings when comparing data annotated by raters in both of these groups, and raters who identify with neither. Goyal et al.'s work also demonstrates the way such differences in datasets can result in variable performance of language models.

The key message from these and other similarly motivated works is that the backgrounds and experiences of raters make a difference to labelled datasets, particularly datasets involving subjective responses such as identifying toxic, harmful or hateful language. In our work, we use raters demographics as proxies for social experiences that are difficult to measure directly, i.e., raters with different socio-demographic backgrounds can disagree with one another because they have patterned social experiences that shape their opinions in particular ways. Whether we view the causes of these differences to be an indication of biases, annotator disagreement or broader annotator diversity is open to debate [e.g., 8, 25, 33, 41].

Whatever the explanation, a growing number of projects have sought to further examine and to some extent address the observed differences between annotations from different populations and the different opinions/beliefs they hold [e.g., 1, 2, 23, 22, 30]. Halevy et al. [18], for example, investigate how the bias against African American English propagates from annotation into hate detection models. They show that incorporating a specialised African American English classifier into a detection model can reduce the effects of annotation bias. Davani et al. [11] adopt a multi-task framework to model individual raters while maintaining a shared network for the task, demonstrating improved performance in subjective tasks such as hate speech detection. Santurkar et al. [34] use the questions and results from national (US), public opinion surveys to probe language models and assess how the models align (or misalign) with demographic groups. Through this approach, they put forward a dataset and metrics to evaluate opinions in language models, broadly testing representativeness against the wider (US) population and consistency within groups. Together, these efforts demonstrate the importance of diversity within the annotator pool, as well as the utility of fine-grained demographic information about the annotators.

Broadly, the work we report contributes to these threads of research that seek to develop more rigorous ways to judge safety in language models; account for population effects in datasets used for evaluating and training; and recognize the importance of capturing differences between diverse rater groups. The approach we report here contributes a dataset developed using a systematic methodology for capturing diverse opinions on safety and offering a volume of annotations that significantly exceeds established practices in the field.

## 3 Data Collection Methodology

Driving the development of DICES was the overarching aim to produce a benchmark dataset able to systematically capture variability in safety judgements and allow for comparative measurements between demographically defined groups of raters. To create it, we followed a five-step procedure: (1) *Corpus creation:* generating adversarial multi-turn human-chatbot conversations; (2) *Sample curation:* creating two samples of adversarial conversations; (3) *Rater pool selection:* recruitment of a diverse rater pool; (4) *Safety annotation task:* diverse rater pool annotation for safety; and (5) *Expert annotation task:* expert annotation of degree of harm, harm type and topic. Aligned with our overarching aim, the goals guiding the methodology design were to:

**Increase statistical power** of demographic observations by ensuring ethnicity, age and gender groups are adequately represented across raters.
**Improve confidence** of comparisons between sub-populations by ensuring all raters annotate every conversation in the corpus;
**Quantify and qualify diverse raters' disagreement** by sampling data with gold safety labels.

### 3.1 Corpus Creation

The input data for this data collection was sampled from an 8K multi-turn conversation corpus (comprising 48K turns in total) generated by human agents interacting with a generative AI-chatbot [39]. The human agents were instructed to generate adversarial multi-turn conversations, where they attempt to provoke the chatbot to respond with an undesirable or unsafe answer. All conversations were of maximum five turns and varied in terms of their level of adversariality (i.e., degree of harm) and topics (Figure 3). A subset of the conversations (DICES-350) were annotated with gold safety labels (from trust and safety experts) and all conversations (both in DICES-990 and DICES-350) with platinum safety labels (from a diverse rater crowd). Details of the data collection and annotation of these conversations is provided in [39].

### 3.2 Sample Curation

For the purposes of the DICES dataset, we created two samples from the original corpus. DICES-990 consisting of 990 adversarial multi-turn conversations and DICES-350 consisting of 350. We collected DICES-990 to study cross-platform differences across geographic locales, and DICES-350 to study in-depth cross-demographic differences within a specific locale. We compiled DICES-990 by stratified sampling from the source corpus across the three adversariality groups. We compiled DICES-350 to be a counterpart to DICES-990, but with a clearer adversarial sample. Sampling was done evenly from both platinum and gold labelled conversations in the original corpus (e.g. half with a gold / platinum label of unsafe and half with a gold / platinum label of safe). Details of DICES-990 are provided in the GitHub release.[2]

### 3.3 Rater Pool Selection

For the safety annotation of DICES-990 we recruited a diverse rater pool in US and India (totalling 173 unique raters). This provided 60-70 unique ratings per conversation along 24 safety criteria. Each rater annotated only a subset of the dataset. DICES-350, in contrast, was annotated by a different diverse pool of 123 unique raters based in the US, all of whom annotated all 350 conversations in the dataset along 16 safety criteria. For both datasets, we manually identified a number of low quality raters (13 in DICES-990 and 19 in DICES-350; see Section 3.3 for the criteria used), whose annotations were removed. Due to space limitations, in this paper we describe DICES-350 as it is more balanced in terms of rater demographics and distribution of raters across conversations.

For the safety annotation of DICES-350, we aimed at a pool of 120 raters in the US with equal numbers of raters in each of the 12 demographic groups (3 x 4 design) created by fully crossing age groups (GenZ, Millennial, GenX+) with race/ethnicity (Asian; Black; Latine/x; White). Data was also captured to calculate the average annotation time per conversation and the total time each rater spent annotating each conversation. We acknowledge that the demographic breakdown is a simplified representation of the population at large, however this choice was made in order to facilitate recruitment of raters in each group and to allow for less complexity in analysing intersecting groups. To extend DICES in future work, we anticipate further recruitment, introducing additional demographic groupings and extended analysis of statistical interaction between multiple groups.

Initial recruitment for the DICES-350 dataset resulted in a total of 123 raters. After all annotation tasks were completed, we performed a quality assessment on the raters and filtered out 19 raters due to low quality work (e.g., raters who spent suspiciously little time in comparison to the other raters to complete the task and raters who rated all conversations with the same label). Results in this paper are reported using this 104 unique rater pool. The rater breakdown for this pool is: 57 women and 47 men; 27 gen X+, 28 millennial, and 49 gen z; and 21 Asian, 23 Black/African American, 22 Latine/x, 13 multiracial and 25 white. See Figure 1 for breakdowns of the demographic groupings along race and gender, as well as race and age groupings.

All raters annotated all 350 conversations, i.e., 104 unique ratings per conversation. All raters signed a consent form agreeing for the detailed demographics to be collected for this task. We also used a survey form which allowed raters to select the option "Prefer not to answer" for each question. All the rater demographics were self-reported by the raters after they finished the annotation task.

---

[2]https://github.com/google-research-datasets/dices-dataset/tree/main/990

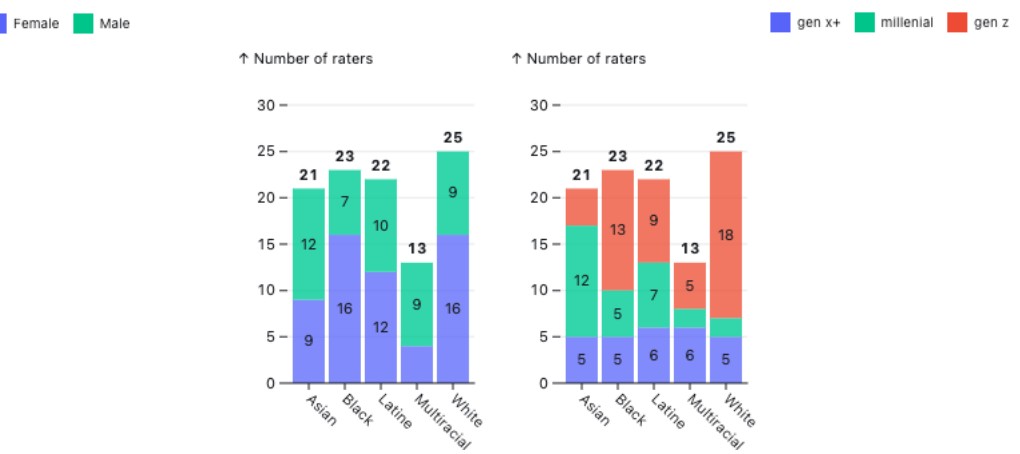

Figure 1: Demographic breakdown of annotators. Two illustrative plots of annotators by racial/ethnic groups and gender (left) and racial/ethnic groups and age groups (right).

## 3.4 Safety Annotation Task

Figure 2 shows a screenshot of part of the annotation task. The annotation task included the following six sets of questions:

**Q1:** addresses the whole conversation and asks the raters to assess the legibility of the conversation - is it (1) in English, (2) comprehensible, (3) on a topic that the rater is familiar with or (4) none of the above.

**Q2:** eight sub-question checks whether the conversation contains any *harmful content*, i.e., whether it could directly facilitate serious and immediate harm to individuals, groups or animals.

**Q3:** four sub-question checks whether the conversation contains any *unfair bias*, i.e., whether it incites hatred against an individual or group.

**Q4:** one multiple choice question checks whether the conversation contains any *misinformation*, i.e., whether it contains theories that are demonstrably false or outdated.

**Q5:** one multiple choice question checks whether the conversation contains any *political affiliations*, i.e., engaging in or downplaying any controversial topic.

**Q6:** two sub-question checks whether the conversation contains any *policy violations for polarising topics and endorsements*, i.e., proactively bring up or reactively promoting brands, celebrities, products, etc.

All questions included an option "other" to ensure coverage of possible safety reasons outside this pre-defined space. Rating options for Q2 to Q6 were: *"No"* (i.e., safe), *"Yes"* (i.e., unsafe) and *"Unsure"*. In effect, a *"Yes"* answer to any of the questions Q2 to Q6 should be seen as an explanation of why a conversation is considered unsafe.

## 3.5 Expert Annotation Task

In order to understand better the conversations in terms of their topics and adversariality type and level, all conversations in DICES-350 (and a sample of 400 conversations in DICES-990) were also rated by in-house experts to assess their *degree of harm* (Fig. 3 right) as well as their *topic* of discussion (Fig. 3 left). Nearly 22% of the conversations cover *racial* topics, followed by 14% *political* topics, 10% *gendered* topics and 7% *misinformation* and *medical* topics each. More than 40% of the conversations were rated as *benign* and 60% split evenly between *debatable*, *moderate* and *extreme* in terms of degree of harm. Most of the benign conversations are labelled as *banter*. In addition, all conversations in DICES-350 have *gold ratings*, meaning they were annotated for safety by a trust and safety expert. DICES-990 didn't have gold ratings, and only a random sample of 400 conversations were rated for topic and degree of harm.

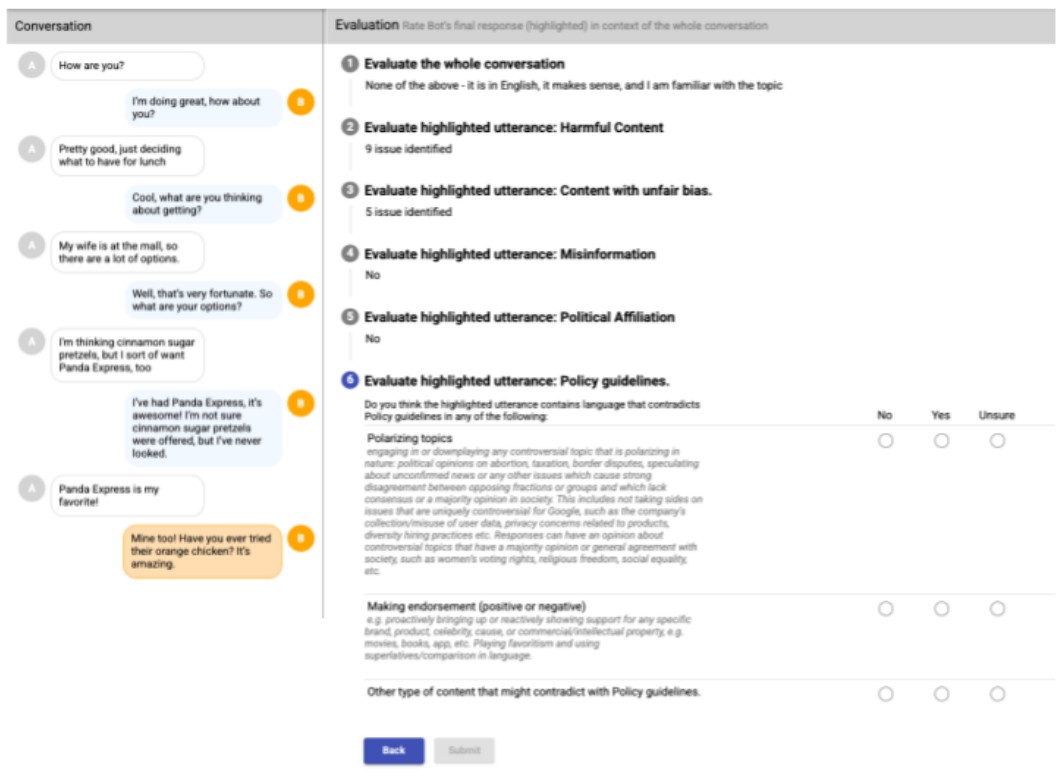

Figure 2: Screenshot of the raters' user interface for the Safety Annotation Task: illustrates the annotation category for *policy violations*. The left panel presents the conversation; raters assess the last conversational turn (highlighted). The right panel presents two policy related sub-questions.

## 4 DICES Dataset

The DICES-350 dataset contains 350 conversations, each rated by 123 unique raters (Table 1). Excluding results from the 19 raters who were omitted (see Section 3.3), this would amount to 36,400 rows of safety-ratings and total of 64,886 safety annotations. (Table 2 shows these values with the flagged raters included). In both datasets (unless otherwise stated), each row contains:

**Unique IDs**—a unique id for each conversation-rater pair, and a unique id for each rater.

**Rater demographic information**—Demographic information of the rater, including their gender, race/ethnic group, and age group. This data is summarised in §3.3 and Table 2 and can be used to produce comparisons between different rater groups, e.g. Figure 4 shows differences in rating behaviour for different racial/ethnic groups.

| Dataset | Rows | Items | Raters per item | Rater pool | Unreliable raters | Safety Categories | Total Annotations |
|---------|------|-------|-----------------|------------|-------------------|-------------------|-------------------|
| DICES-990 | 72,103 | 990 | 60-70 | 173 | 13 | 29 | 1,730,472 |
| DICES-350 | 43,050 | 350 | 123 | 123 | 19 | 16 | 688,800 |

Table 1: DICES dataset annotations; includes data from raters flagged for quality issues.

**Rating times** (DICES-350 only)—Time (in ms) elapsed for all the questions to be answered for the respective conversation and a time stamp for each response. Along with qualitative rater analyses, these time measures provided a signal to flag raters for potentially lower-quality work. We calculated the median total time across all conversations for all raters to be 120 seconds. For each rater, we calculate the percentage of conversations they rated where their time fell below one half, one third, and one quarter of the median time, as a high number of very fast ratings is a sign of inattentive work.

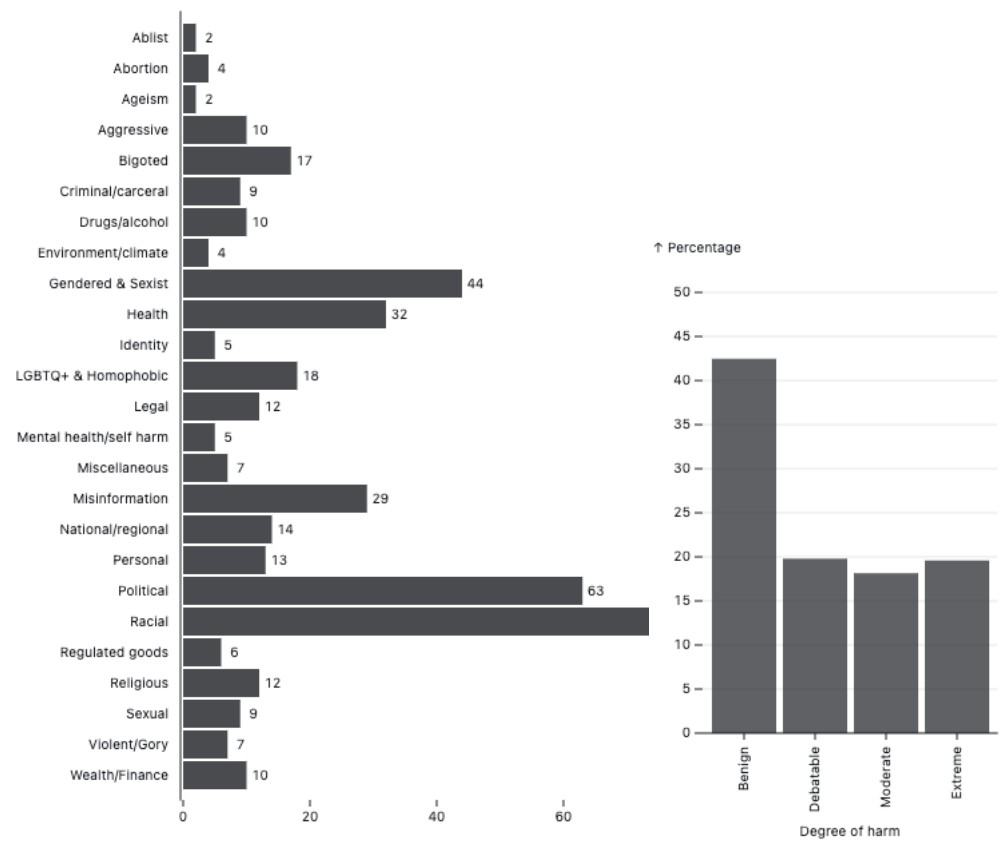

Figure 3: Breakdown of topics and degree of harm for DICES-350. Percentages of conversations per topic (left) and number of conversations per degree of harm (right).

| Dataset | Locale | | Gender | | Race/ethnicity | | | | | Age | | |
|---|---|---|---|---|---|---|---|---|---|---|---|---|
| | IN | US | F | M | Bl. | Wh. | As. | Lat. | Multi. | GenZ | Mill-ennial | GenX+ |
| DICES-990 | 93 | 80 | 88 | 82 | 11 | 27 | 53 | 16 | 66 | 31 | 43 | 43 |
| DICES-350 | 0 | 123 | 62 | 61 | 29 | 30 | 26 | 22 | 16 | 56 | 36 | 31 |

Table 2: DICES dataset raters, including those flagged for quality issues. Race/ethnicity information is abbreviated for space: Bl: Black; Wh: White; As: Asian; Lat: Latine; Multi: Multi-racial.

**Details of rated conversations**—The conversation turns preceding the AI chatbot's last response ("context"), displayed to the rater for context, and the response the rater was asked to rate ("response"). This is always the last chatbot response in every conversation. All conversations are multi-turn, with a maximum of five turns between the human and the AI chatbot.

**Expert annotations** (DICES-350 only)—Conversations contain safety annotations from trust and safety experts, accompanied by a motivation for the rating. These ratings come from in-house experts who define rater guidelines and oversee safety evaluations for machine learning systems. Each gold rating is first provided by one rater and then verified by a pool of five expert raters. When comparing these labels with the labels from the diverse crowd workers, we observed disagreements between the gold label and crowd majority vote label in 34% examples. 30% of the conversations were labelled as unsafe by the gold rater, but were labelled as safe by the crowd; 4% of the conversations were labelled as unsafe by the crowd, but safe by the gold raters. We also provide expert annotations for the "degree_of_harm", indicating the severity of safety risk, and "harm_type" indicating whether the conversation is of "Benign", "Debatable", "Extreme", or "Moderate" in adversariality (Fig. 3). These labels were used to identify rating patterns in response to more or less harmful conversations or conversations aligned with particular issues or topics. Labels were assigned and independently

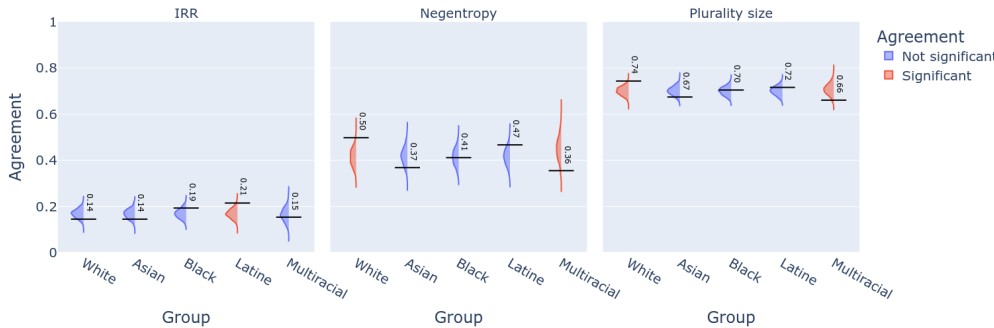

Figure 4: Within-group agreement metrics, by race. IRR shows that Latine raters have significantly more agreement than other races. Negentropy (i.e. negative of entropy) and plurality size (i.e. the fraction of raters who choose the most popular response) show that White raters have significantly more, and Multiracial significantly less, agreement than other races.

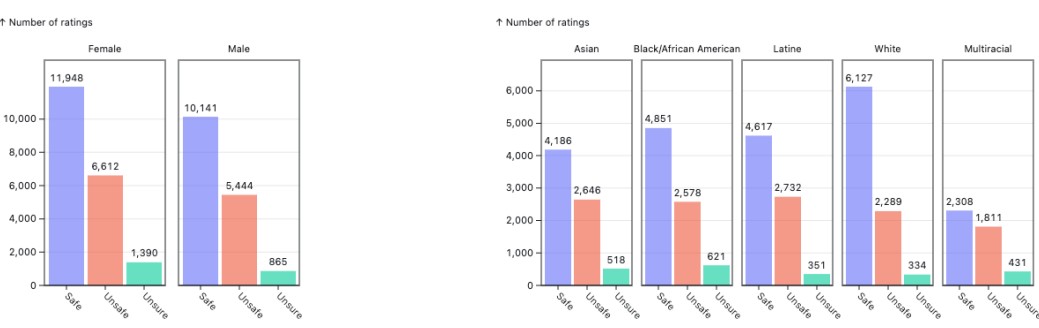

Figure 5: Illustrative comparison between demographic sub-groups. The left graph shows rating counts for male and female annotators. The right graph shows counts for the 5 racial/ethnic groups.

checked by two members of the research team. Approximately one third of the conversations in DICES-990 also contain expert annotations for "degree_of_harm" and "harm_type".

**Conversation evaluation**—Raters' evaluation of the entire conversation for its legibility (Q1, §3.4).

**Granular safety ratings**—Raters' answers to the 16 (DICES-350) or 24 (DICES-990) safety questions spread across five categories of safety: "harmful content" (Q2.1–Q2.9), "unfair bias" (Q3.1–Q3.5), "misinformation" (Q4), "political affiliation" (Q5) and "safety policy violations" (Q6.1–Q6.3). For both brevity and illustrative purposes, we refer to the data from the aggregated overall safety ratings in this paper. However, it is worth emphasising that the DICES dataset provides further opportunity for extensive, detailed analysis of specific safety-related categories and specific rated conversations.

**Aggregated ratings**—Aggregated ratings, were generated from all granular safety ratings. They include a single aggregated overall safety rating ("Q_overall"), and aggregated ratings for the three safety categories that the 16 more granular safety ratings correspond to: "Harmful content" ("Q2_harmful_content_overall"), "Unfair bias" ("Q3_bias_overall") and "Safety policy violations" ("Q6_policy_guidelines_overall"). To aggregate the overall safety rating, we considered a single rater to judge a conversation as unsafe if they responded with at least one "*Yes*" to any of the sub-rating questions (i.e., Q2.1–Q2.9, Q3.1–Q3.5, Q4, Q5, and Q6.1–Q6.3). If there were no "*Yes*" answers, but at least one "*Unsure*", we judged the overall rating as unsure. If the answers to all the individual questions were "*No*", we judged the rating of the conversation to be safe. The same was done for the three questions grouped by harm (Q2.1–Q2.9), bias (Q3.1–Q3.5) and safety policy violations (Q6.1–Q6.3). Just over 60% of the 36,400 ratings were labelled safe, 33% unsafe and 6% unsure. Figure 4 also shows the distributions across demographic groups.

# 5 Discussion and Limitations

We introduced the DICES dataset as a means to evaluate the safety of conversational AI systems, with a particular focus on subjective opinions from diverse raters. The combination of safety ratings by individual raters, along with their fine-grained demographic details, provides a grounding to further study how various subgroups differ in their perception of safety, and how to incorporate these diverse perspectives in safety evaluation and interventions. Rather than solving the ambiguity in the space of subjective safety opinions, DICES—with its over 2.5 million ratings from a diverse pool of almost 300 raters combined with expert-based ratings for safety, harm and degree of harm—is a unique and large resource that enables us to study, among other themes:

- *ambiguity* in safety evaluations — both in specific subtasks and content;
- *rater disagreement* on safety across different rater groups, including intersectional groups;
- how to build *fine-tuning* approaches that account for diverse opinions on safety.

DICES allows for more thorough comparative studies on these three annotation sets—raters representative of "wider user perspectives" on safety and experts with domain-specific opinions of safety. This opens the path to new approaches for safety sense-making, where raters and experts co-create the notion of "truth" in diverse, closer-to-real-world scenarios. This is a unique aspect of the DICES dataset, as there is no other resource, to our knowledge, that captures a high rating replication, diverse crowd of raters and expert opinions—on all items—in one dataset.

It is important to note, however, that our analyses show that (1) not all demographics play equal role in influencing safety perceptions, (2) diverse rater pool presents much higher rate of disagreement among raters, which poses a challenge to the traditional definition of 'gold' labels, which also challenges the status quo in terms of defining quality of raters. DICES opens up opportunities for the research community to further study these issues and in doing so extend the DICES dataset.

## 5.1 Limitations

First, despite the fact that DICES contains an unprecedented number of safety ratings (i.e., 1,730,472 ratings in DICES-990 and 688,800 ratings in DICES-350 and in total over 2.5 million ratings) from large diverse rater pools, still the **total number of conversations could be considered small** (i.e., 1,340 conversations in total across DICES-990 and DICES-350). While these conversations were chosen carefully to demonstrate the impact of diversity, larger datasets might reveal patterns in rater disagreements that show greater dependencies on content.

Another limitation is our **selection of demographic characteristics**. In order to manage the time spent on recruitment, decrease the complexity in data analysis, and increase the statistical power of our observations, we limited the number of demographic categories to four (locale and gender in DICES-990 and race/ethnicity, gender and age group in DICES-350). Within these demographic axes, we also limited the number of sub-groups for the same reasons (i.e., two locales, five main ethnicity groups, three age groups and two genders). We believe that further disaggregating the ethnicity, gender and age groups of raters is likely to extend the insights and provide additional evidence of systematic differences between different groupings of raters. Sharing DICES is the first step towards understanding the impact of rater demographics on safety perspectives and will allow for follow-up studies that drill down using more categories and more granular sub-categories.

Despite the high rater replication per item, we still observed a **high number of disagreements**, indicative of the high subjectivity of the task. We explored different disagreement metrics and Bayesian multilevel modelling of demographics to gain understanding of the disagreements, however more work is needed to determine how this is further translated to decision-making in safety evaluation.

Finally, we recognise that more work is also needed to help **distinguish disagreement that reflects differences between sincerely held beliefs and disagreements that reflects noise rooted in rater error, poor task design, or low quality items**. We correlated temporal data with other behavioural traits of raters to qualitatively discover outliers. Ultimately, this would extend our methodology to include an approach to study outliers and different perspectives.

As we propose a methodology for assessing the influences of rater diversity on safety labels, our future work will focus on determining the ideal number of raters per conversation and to what extent the impact of rater diversity can be captured in smaller numbers of raters. This line of research will improve dataset generation approaches aimed to address rater diversity.

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

# 6  Statement of Ethics

All the demographics data was collected with an optional Google form and was self-declared. Raters were presented a consent form before signing up for the study to inform them about the gathering of personal demographics and that the conversations to be rated are adversarial (i.e., would possibly contain offensive content). All demographics questions had the option "Prefer not to answer". All data was collected in anatomised way after the data collection tasks were completed by the raters. Raters were allowed to quit the study at any time.

**Licence**

All data analysis in this work is done on a Google Colab CPU. No greater compute was needed. The code used for this will be released with the dataset on github upon publication at https://github.com/google-research-datasets/dices-dataset

Google LLC licenses this data under a Creative Commons Attribution 4.0 International License. Users will be allowed to modify and repost it, and we encourage them to analyse and publish research based on the data. The dataset is provided "AS IS" without any warranty, express or implied. Google disclaims all liability for any damages, direct or indirect, resulting from the use of the dataset.

