# OpenReview forum: "DICES Dataset: Diversity in Conversational AI Evaluation for Safety"
_NeurIPS.cc/2023/Track/Datasets_and_Benchmarks — NeurIPS 2023 Datasets and Benchmarks Poster_

### Official Review · Reviewer_HcmN · 2023-07-21
**A Powerful Entry in the Data Evaluation Space**

**Rating:** 9
**Confidence:** 4
**Correctness:** Yes
**Clarity:** Yes

**Strengths:**

The paper does a great job at addressing many concerns that have been flagged around annotation based datasets and takes great care to ensure they do not stumble into the same "traps". They are very thoughtful with the demographics of their annotators and providing the public with a breakdown of how raters rate conversations across groups, how safety is defined across groups and showing that there can be some ambiguity around what constitutes a harm- is the harm benign, debatable, or extreme? Does everyone agree? How do these differences present themselves and does these differences mean better work should be going into defining terms or are some things simply subjective. In the end, the researchers acknowledge that tension and the subjectivity around these tasks. They do not shy away from conversations around demographic difference and how different groups may disagree with what they consider to be harmful.

**Additional Feedback:**

Add the metrics and results discussed on page 2 of the paper. If this is not going to be included, then perhaps removing that line may be best.

**Documentation:**

While the data is available in the git repo shared as a part of this submission, I do not see where the metrics and results of the dataset evaluation are. On page 2 of the paper, the authors say that these metrics will be available within the dataset repo so I would like to see that included.

**Limitations:**

Yes, the authors state their limitations as being subjectivity of the annotations; larger than before but still small datasets; a relatively small representation of "diversity" as demographic subgroups were restricted to age, gender and race; and lastly, a lack of distinction between disagreement between annotators that is rooted in true difference or simply noise. These are notable limitations and it is good to include them. However, there is another big limitation here is the bias toward native English speakers. It would be good for the authors to address whether some conversations were had by people who do not speak English as a first language could cause material differences in both how their conversations may be rated as well as how they may rate conversations. There is one measure called "Comprehensible" that I imagine could have some bias against folks who speak English as a second or third language. Considering African American English use is addressed to reduce bias, which is amazing by the way, it feels appropriate to include this measure as well- at least as a thing to include in the future.

**Opportunities For Improvement:**

It could be useful to consider, if the data is available, that in-groups may be most equipped to define what is harmful to them and see if the difference in what is considered harmful across groups is greater or lessened based on the group being harmed. For example, Asian people would be best equipped to define what is harmful when directed toward them, women would be the best judge of what would be sexist, older folks would be most aware of ageist language, etc. It would be interesting to see if differences are at their most stark when comparing the rating of an in-group by groups that historically been juxtaposed to them. So for example, are younger people more likely to disagree with an older demographic about what is ageist, are men rating sexist language less often than women, etc.

**Relation To Prior Work:**

Yes

**Summary And Contributions:**

The Dices Dataset is composed of two annotated datasets of human-bot conversations. One dataset with 990 conversations with 70 annotations per conversation and the other has 350 conversations with 120 annotations per conversation. The authors consider work done by Halevy et al. that shows that annotated work in the past has carried with it biases against African American English speakers and have gone through lengths to add classifiers as proposed by Halevy et al as well as other measures like diversifying the annotators themselves to be representative of the population. The authors also consider different definitions of safety and include a wider definition of the term that includes harms, biases and misinformation. Lastly, they include metrics of effectiveness for researchers to use for dataset evaluation.

---

> ### Author Response · Authors · 2023-08-22
> **Response to review by HcmN**
>
> We thank the reviewer for their helpful feedback and recommendations. We address the reviewer’s concerns below.
>
> 1. It could be useful to consider, that in-groups may be most equipped to define what is harmful to them and see if the difference in what is considered harmful across groups is greater or lessened based on the group being harmed.”
>
> We did perform explorations (both qualitative and quantitative) on this question, however, they are presented in a different paper (currently under review as anonymous submission, thus not able to share at this moment unfortunately). To bring some clarity on this here, we summarize some of the high level findings here (also more details are now included in Appendix 7):
>
> – Over the entire dataset, only moderate associations with rater behavior where observed between race and age independently
> – Intersectional effects between race and gender were strongly associated with rater behavior, for example white men being significantly more likely than other raters to rate conversations as “safe”.
> – Younger raters were more likely to rate conversations as safe, and we observed a strong correlation between age and sensitivity to “safety”
>
> 2. Dataset is  small; a lack of distinction between disagreement between annotators that is rooted in true difference or simply noise.
> The actual numbers of unique items in the DICES dataset is indeed small. However, this dataset is not intended for training, but rather for evaluation:
>
> -- Evaluation sets are typically small in terms of number of items. Current approaches for evaluation of conversational AI systems typically use datasets between 1000-3000 examples. DICES contains 1340 examples in total. As it is intended to provide a resource for studying the impact of rater demographics we focused on increasing the number of raters for each example and allow for statistical analysis with significance.
>
> -- DICES dataset is of an unprecedented size in terms of human annotations - it presents 2,5 million safety annotations distributed across 4 main demographics and their subgroups) with a significantly larger number of ratings per item (by unique raters) than equivalent datasets. As such DICES is a valuable resource for statistical analysis of demographic diversity and drawing out statistical comparisons to gold ratings. These annotations enable a richer understanding of model safety. Specifically, DICES allows for judging, (i), the extent to which LLMs capture demographic diversity when evaluating their safety and, (ii), how well they capture diversity/disagreement with gold safety evaluations. There is currently no match for such a dataset available for the evaluation of safety of conversational LLMs. We are not aware of any other safety evaluation datasets of this scale in terms of number of parallel annotations.
>
> -- DICES is also intended to be a “living” dataset - this is the first version of DICES and we plan to continuously update it over time with more granular representation of demographics and new examples over time to increase its evaluation power and relevance to the current generative AI models status quo. We also included some clarification in the discussion section (line 336).
>
> We also included in the limitation / future work section a statement about distinguishing between disagreements rooted in demographics differences and simply noise or low quality raters (line 336). In an additional paper (currently in anonymous review period) we performed both qualitative and quantitative analysis on rater quality to clarify this distinction. In summary, only 12 to 19 raters in both datasets (from a pool of more than 100) have been flagged as low quality contributors. All the disagreement analysis was performed on the data excluding these raters in order to only observe the demographic differences.
>
> 3. “Bias toward native English speakers.”
>
> Thank you for bringing up the potential relationship between non-native speakers and comprehensibility. Indeed, this dataset has a strong bias towards English speaking raters - the conversations we generated in a red-teaming effort with a mixture of people with different levels of English comprehension as their native or second language - all with high English proficiency (daily use at work). We agree that, this is a good distinction to explore further, thus in the limitations / discussion / future work section we added a point for the need for such exploration by including it in the demographics survey for conversation generation teams (similar to the one we used for the raters of these conversations, see appendix 3, and adding a question like Question 1 illustrated in Appendix 4).
>
> 4. “I do not see where the metrics and results of the dataset evaluation are”
>
> We do have a publication under review (in anonymity period, thus not able to share at the current moment) focusing on metrics used for this evaluation. In order to bring some clarity on this here we included details in Appendix 7

---

### Official Review · Reviewer_45Pu · 2023-07-21

**Rating:** 6
**Confidence:** 4
**Correctness:** Seems correct overall.
**Clarity:** It was clear to read.

**Strengths:**

- The research acknowledges and addresses issues of subjective bias in the evaluation and safety criteria within AI systems. It takes into consideration diversity and generates a nuanced approach towards understanding safety measures, rather than simplifying or negating inherent subjectivity.
- The authors used a broad selection of raters from different demographics, which facilitates a comprehensive exploration of potential bias. The expansive pool helps ensure reliable results and measurement of diversity, differences in safety perceptions and the potential effects of these differences on ratings of Conversational AI for safety.

**Additional Feedback:**

None.

**Documentation:**

Yes, I see the required documentation mentioned in the submission.

**Ethics:**

Since the paper involves highly sensitive data, I would flag the paper for an ethics review as datasets with such level of data could be easily miss-used.

**Limitations:**

While the authors mention that they used raters in the U.S and India for DICES-990 and only in the U.S for DICES-350, these two countries represent only a small portion of global perspectives. There is a risk of cultural bias characterizing what is 'safe' or 'unsafe.' Inclusion of raters from a broader range of geographic locations and linguistic backgrounds could strengthen the dataset and offer a more comprehensive view of safe conversational AI practices globally.

**Opportunities For Improvement:**

- The research provides a focus on cultural, social, and demographic diversity but does not discuss the impact of individual personality traits or personal experiences beyond demographic segmentation. These factors can contribute considerably to one's perception of safety and should be taken into account when constructing such datasets.
- The demographic breakdown may be oversimplified, thus omitting nuances within demographic groups. The division into three age groups (GenZ, Millennial, GenX+) and four ethnic/racial categories may miss more granular differences within these broad categories. While the breakdown was chosen to reduce complexity in the analysis, it may limit the insights that could be gained from more nuanced demographic data.

**Relation To Prior Work:**

Yes, the paper covers the previous works in this area well.

**Summary And Contributions:**


The paper introduces Diversity in Conversational AI Evaluation for Safety (DICES), a dataset aimed at measuring and understanding variance, ambiguity, and diversity in the safety evaluation of conversational AI systems. Recognizing the inherent subjectivity in safety evaluations, DICES leverages diverse demographic information about raters, high replication of ratings per conversation and the encoding of rater votes for robust analysis. The authors designed DICES to serve as an open resource for safety evaluations of conversational AI system, allowing for an intersectional examination of raters' ratings with demographic categories such as racial/ethnic groups, age groups, and genders.

---

> ### Author Response · Authors · 2023-08-22
> **Response to review by 45Pu**
>
> We thank the reviewer for their helpful feedback and recommendations. We address the reviewer’s concerns below.
> 1. “The research does not discuss the impact of individual personality traits or personal experiences beyond demographic segmentation.”
>
> This is an important point and we acknowledge the analyses presented in this paper do not account for individual differences in this way. We do note the subjective nature of this task, and mention that more work needs to be done to better understand decision making as it relates to a wide range of individual and social factors. In fact, we have a parallel work under submission where we perform a qualitative analysis on individual rater responses  to discover behavior patterns and align these with personal preferences and attitudes towards safety.
>
> 2. “The demographic breakdown may be oversimplified, thus omitting nuances within demographic groups."
>
> We are acutely aware that the demographic groups we used thus far are simplistic as stated in section 3.3 “We acknowledge that the demographic breakdown is a simplified representation of the population at large, however this choice was made in order to facilitate recruitment of raters in each group and to allow for less complexity in analyzing intersecting groups” (line 178). Exploring the effects of specific demographic groups, or intersections of demographic groups requires a large coverage of raters from different demographic groups. For instance, even with 120 raters per item, currently there are only 9 white men represented in the DICES-350 dataset. The simplified demographic distributions (derived from a power analysis to determine an optimal size of the rater pool for statistical power within each demographic group) allowed for increasing the number of annotators in each top-level demographic group and allowed for this initial in-depth demographics analysis. As a result we can present strong initial indication of how diversity impacts safety ratings and how diversity could be better accommodated in future work.
>
> The DICES dataset is intended to be a “living” dataset - the presented version is to be extended (both in size and nuance of the demographic categories) by ourselves and the wider research community. We see this early access to the dataset as critical in order to understand how others would use the dataset and to derive requirements to develop, extend and refine it further. We plan to continuously keep updating it over time with more granular representation of demographics as well as with new conversations to increase both its size and its evaluation power and ultimately its relevance to the generative AI models status quo as it changes. We also included some clarification in the discussion section (line 336).
>
> 3. “Raters in the U.S and India represent only a small portion of global perspectives. There is a risk of cultural bias characterizing what is 'safe' or 'unsafe.'”
>
> We do acknowledge this as a limitation of the dataset as it currently stands (also included in the limitation section of the paper). Because of the cultural specificity of certain demographic characteristics (e.g., race/ethnicity), the ability to compare demographic information becomes more limited as the scope of inclusion expands. As a result, we made a trade-off to first optimize for depth of analyses within a more limited scope, and plan for future expansions that focus on more global characteristics (e.g. including raters from different countries and cultural backgrounds) and also statistical analysis to evaluate what effect this has on safety ratings.
>
> 4. “I would flag the paper for an ethics review as datasets with such a level of data could be easily miss-used.”
>
> We agree that this dataset could potentially introduce the novel harm of allowing someone to create models more biased in the eyes of one particular demographic. This is, however, the case of all publicly available adversarial datasets, especially in the field of fairness and bias. However, at the same time, making this dataset publicly available also allows researchers to improve model safety by uncovering this kind of unintended bias that may emerge as a result of not evaluating rater demographics and social groups at all. We believe that it is very unlikely someone will *accidentally* use our dataset to create a more biased model as the dataset would require significant manipulation to be able to be used as is in fine tuning. Stopping intentional malicious use of the dataset is much more challenging and unpredictable. In this case we believe the pros of providing this resource to the research community outweigh the cons. We are publishing this dataset to increase awareness of demographic variables in safety considerations, which we hope will spur the research community towards better mitigation against even intentional misuse of datasets like ours.

---

> > ### Comment · Reviewer_45Pu · 2023-08-23
> >
> > Thank you for the response. I still don't find the reasoning for the weaknesses convincing enough to change the score, and will keep the same score of 6.
> >
> > Also, I would like the authors to cite some of the benchmarks related to AI abilities such BOLD[1] and ToxiGen[2] as it is important for readers to stay aware about some of the risks posed by AI when their abilities are improved.
> >
> > **References**
> >
> > [1] Dhamala, Jwala, et al. "Bold: Dataset and metrics for measuring biases in open-ended language generation." Proceedings of the 2021 ACM conference on fairness, accountability, and transparency. 2021.
> >
> > [2] Hartvigsen, Thomas, et al. "Toxigen: A large-scale machine-generated dataset for adversarial and implicit hate speech detection." arXiv preprint arXiv:2203.09509 (2022).

---

> > > ### Author Response · Authors · 2023-08-28
> > >
> > > Thank you for reading through our responses. Below you can find some clarifications:
> > >
> > > “Also, I would like the authors to cite some of the benchmarks related to AI abilities such BOLD[1] and ToxiGen[2] as it is important for readers to stay aware about some of the risks posed by AI when their abilities are improved.”
> > >
> > > Thank you for suggesting the two benchmark references. We did examine them carefully, and included them in our related work (current version of the paper is updated). We acknowledge that both papers are indeed relevant as they release two datasets (one for toxicity and one for bias). However neither of them releases any human ratings. The BOLD dataset paper discusses a limited human validation task with just 5 raters per item. Toxigen provides no detail on the actual human annotation task. In the case of DICES dataset, we do release all granular human ratings (upto 120 ratings per item). We show empirically how the DICES dataset is offering a unique utility for in-depth statistical and qualitative studies of diversity in this space (with an unprecedentedly high number of rating) and are complementarity to current state of the art efforts.
> > >
> > > “I still don't find the reasoning for the weaknesses convincing enough to change the score”
> > > It would be really great if the reviewer could clarify which (if not all) of the perceived weaknesses they feel were not addressed sufficiently in our rebuttal. This will help us to focus our revisions on them. Below we iterate on our perception of the value of the DICES dataset:
> > >
> > > -- the main point of this dataset is to meet the urgent need in the Generative AI field to gain a deeper understanding of diversity implications on safety. Collecting this dataset with the selected (though simplified) demographics setup took over 6 months (from rater recruitment along all the simplified demographics categories to achieve optimal coverage, survey construction, data annotation to finally ethics evaluation of the resulting data). Addressing all complexity of demographics and locales in an experimental setup from the start would have increased this time frame exponentially, and in some cases would have made it impossible to do so from the start. Given the speed with which the Generative AI field is moving, it is more important to start with a scoped understanding than to attempt to capture all aspects of individual differences right from the start.
> > >
> > > -- The main purpose of the DICES dataset is to meet the urgent need in the Generative AI field to gain a deeper understanding of diversity implications on safety. it shows empirically that safety perceptions vary depending on a person's demographics and locale, and we believe it is a valuable resource for researchers and developers who are working to make Generative AI safer and more inclusive.
> > >
> > > -- After carefully reviewing the state of the art and related datasets in this field, we believe the DICES is a unique and a valuable addition to the field of Generative AI safety research. It provides a simplified view of the real-world diversity of safety perceptions, which can be used for both statistical and qualitative analysis. While not perfect, is a beneficial first step towards a deeper understanding of diversity implications on safety in Generative AI and towards developing more effective safety interventions. We are committed to continuing to update and expand the dataset, and we hope that you will reconsider your score.
> > >
> > > We hope these points provide sufficient argumentation for the value of this research.

---

### Official Review · Reviewer_aL9d · 2023-07-26
**Dataset that captures ambiguity in safety evaluations**

**Rating:** 7
**Confidence:** 4
**Clarity:** The paper is very well written.

**Strengths:**

The main strength of the paper is the data collection paradigm that paid attention to the ethnic and demographic background of raters. This diverse pool of raters recruited for the study will provide a diverse and a more accurate representation of safety in society instead of the notion of a single ground truth that is obviously not true in many real-world settings.

The second strength of this paper is the number of annotations per conversation (70-120) over (3-5) which is generally the case. This will significantly improve our understanding of each conversation and the statistical reliability from a larger number of ratings will also help us understand current limitations of LLMs.


**Additional Feedback:**

N/A

**Correctness:**

This is a dataset paper and the data was collected with a lot of attention to detail to capture the ambiguity in current safety evaluation protocols.

**Documentation:**

Very well documented pipeline on how the authors collected the data.

**Ethics:**

The authors discuss the dataset collection guidelines and from the available information it does not appear that there are any ethical violations.

**Limitations:**

The authors clearly addressed the demographic issues and the small dataset samples in their work.  The authors also discuss implications of the disagreement they capture through the lens of deeply held beliefs and poor task design.

**Opportunities For Improvement:**

While this approach is costly 350 or 900 are too few to evaluate these LLMs. It would be great to expand these. Also, does it make a significant difference while increasing from 50 annotations to 120? At what point does the gain stop being useful and resources could be better spent in collecting more conversations than annotations?

**Relation To Prior Work:**

The paper discusses the related work in great details and it differs from previous contributions.

**Summary And Contributions:**

With so much contamination of training and test sets and current known issues with large language models, this paper clearly dives into very important and critical aspects of safety that will be very beneficial to the community. The dataset focusses on an important aspect of safety which is the ambiguity because of cultural, demographic and ethnic backgrounds of raters. With such a wide pool of ratings across two different countries (US and India) and multiple raters, this dataset can play a critical role in evaluating diverse topics.

---

> ### Author Response · Authors · 2023-08-22
> **response to review by aL9d**
>
> We thank the reviewer for their helpful feedback and recommendations. We address the reviewer’s concerns below.
>
> 1. “While this approach is costly 350 or 900 are too few to evaluate these LLMs. It would be great to expand these.”
>
> The DICES dataset is intended to be a “living” dataset - the current publication is the first version and we plan to continuously keep updating it over time with more granular representation of demographics as well as with new conversations to increase both its size and its evaluation power and and ultimately its relevance to the generative AI models status quo as it changes. We also included some clarification in the discussion section (line 336).
>
> 2. “Does it make a significant difference while increasing from 50 annotations to 120? Does the gain stop being useful and resources could be better spent in collecting more conversations than annotations?
>
> -- We have performed explorations of evaluating the optimal number of raters, and we published it in paper https://arxiv.org/abs/2306.11530. We also included a detailed explanation of the experiments in Appendix 6 in the supplementary materials. Our results there suggestEstimates for statistics, such as mean values, converge to the actual result at around 20 annotations.
>
> -- However, the variance of these estimates continues to decrease as the number of annotations increases to 120.
>
> -- We anticipate that most users of this dataset will want to explore effects on specific demographic groups, or intersections of demographic groups, which requires a coverage of raters from different demographic groups. For instance, even with 120 raters per item, there are only 9 white men represented in the DICES-350 dataset, so having more future annotators in various granular demographic groups is very helpful for in-depth demographics analysis.

---

### Official Review · Reviewer_jiuK · 2023-08-05

**Rating:** 7
**Confidence:** 3
**Correctness:** Yes.
**Clarity:** Yes.

**Strengths:**

### Strength

This paper investigates human-annotated large language model interaction datasets concerning different races, ages, and education levels. The quality of the datasets was examined to a certain extent, and both the datasets and the production details were made publicly available.

### Weakness

Paper submission guidelines. This paper exhibits several instances of non-compliance with the formatting requirements, ranging from the main body to the images, including but not limited to the absence of line numbers and the presence of blurred images.

Lack of experimental validation: The paper lacks concrete experimental validation to demonstrate the superiority of the proposed method for annotating safety preferences based on different user demographics.

**Additional Feedback:**

N/A

**Documentation:**

N/A

**Ethics:**

No.

**Limitations:**

The dataset size is relatively small; the performance and feasibility of the dataset have not been validated on actual large language model training; the categorization of different user groups is overly simplistic.

**Opportunities For Improvement:**

- Why is the main text lacking the required line numbers for review?
- Why is there a discrepancy in the clarity of images between the main text and the appendix, with the images in the main text, such as Figure 2, being significantly blurry?
- The main text includes an appendix.
- The formatting of the appendix is inconsistent with the main text, and why is the font different as well?
- The second page of the appendix is blank
- The figure captions in the main text are bolded, whereas the table captions are not bolded.
- Does using the proposed classification dataset for training result in better performance, increased safety, or any impact on the original performance of the language model? Furthermore, are there any ablation experimental results(trained LLM) provided?
- The first page's footer contains the term "pre-print."

**Relation To Prior Work:**

Yes.

**Summary And Contributions:**

This paper argues that the safety of language model outputs needs to be differentiated based on different types of individuals due to variations in values resulting from diverse regions, age groups, and knowledge levels. Unlike traditional methods, the safety assessment should not be treated uniformly. The study introduces harm, bias, misinformation, politics, and safety policy violations as metrics and establishes the DICES-350 and DICES-990 human-AI interaction datasets considering variations in ethnicity, age, and geographical regions. The paper evaluates the dataset quality through expert reviews and question-answering time assessments. Ultimately, the analysis examines the level of risks in the dataset and the phenomenon of varying perspectives among different types of annotators. The paper presents a question-answering system for the creation of the Q&A dataset and discloses the recruitment information and composition of data annotators. A preliminary statistical analysis was conducted on the data annotators' sexual orientation, preferences, and educational background.

---

> ### Author Response · Authors · 2023-08-21
> **response to reviewer jiuK**
>
> We thank the reviewer for their helpful feedback and recommendations. We address reviewer’s concerns below:
>
> 1. "Compliance with formatting guidelines & image quality": We have addressed all formatting points (e.g. line numbers, figures not blurred, removed supplementary material from the main text, table captions repaired). Note that the “Preprint. Under review.” is from the latex preprint template.
>
> 2. "Lack of experimental validation to demonstrate the superiority of the proposed method for annotating safety preferences" - We would like to clarify that the safety annotation procedure is not one of the proposed novelties of this paper. For the creation of the DICES dataset we used the same safety annotation procedure (i.e. rater template and safety categories) as defined and validated in https://arxiv.org/abs/2201.08239. The reason to use the same annotation procedure is that we wanted to provide the opportunity to compare diverse ratings with gold expert ratings gathered with an identical annotation process.
>
> 3. "Does using the proposed classification dataset for training result in better performance, increased safety, or any impact on the original performance of the language model? Furthermore, are there any ablation experimental results(trained LLM) provided?" - We would like to clarify that the DICES dataset is not intended as a training dataset, but as an evaluation of safety of conversational LLMs (as stated in abstract (line 18), introduction (line 45)). The original gold ratings from https://arxiv.org/abs/2201.08239 are compared with the diverse safety ratings collected here. We show that in 34% of the dataset conversations there is a clear disagreement between diverse crowd and the gold ratings (line 251). As the number of raters in DICES varies between 70 to 120 raters per item, this allows us to measure statistical significance for all differences observed in our analysis. To further address the reviewer’s point we also included a statement in the discussion and limitations section. (line 333)
>
> 4. "The dataset size is relatively small" - The actual numbers of unique items in the DICES dataset is indeed small. However, this dataset is not intended for training, but rather for evaluation:
>
> -- Evaluation sets are typically small in terms of number of items. Current approaches for evaluation of conversational AI systems typically use datasets between 1000-3000 examples. DICES contains 1340 examples in total. As it is intended to provide a resource for studying the impact of rater demographics we focussed on increasing the number of raters for each example and allow for statistical analysis with significance.
>
> -- DICES dataset is of an unprecedented size in terms of human annotations - it presents 2,5 million safety annotations distributed across 4 main demographic categories and their subgroups) with a significantly larger number of ratings per item (by unique raters) than equivalent datasets. As such DICES is a valuable resource for statistical analysis of demographic diversity and drawing out statistical comparisons to gold ratings. These enable a richer understanding of model safety, e.g. allows for judging, (i), the extent to which LLMs capture demographic diversity when evaluating their safety and, (ii), how well they capture diversity/disagreement with gold safety evaluations. There is currently no match for such a dataset available for the evaluation of safety of conversational LLMs. We are not aware of any other safety evaluation datasets of this scale in terms of number of parallel annotations.
>
> -- DICES is also intended to be a “living” dataset - this is the first version of DICES and we plan to continuously update it over time with more granular representation of demographics and new examples over time to increase its evaluation power and relevance to the current generative AI models status quo. We also included some clarification in the discussion section (line 336).
>
> 5. "The categorization of different user groups is overly simplistic" - We are aware that the demographic groups we used thus far are simplistic as stated in section 3.3 “We acknowledge that the demographic breakdown is a simplified representation of the population at large, however this choice was made in order to facilitate recruitment of raters in each group and to allow for less complexity in analyzing intersecting groups” (line 178). This work represents initial efforts to investigate diversity in annotation thus providing some early indication of how diversity impacts safety ratings and how diversity could be better accommodated in future work. Our intention with the presented dataset is for it to be extended (both in size and nuance of the demographic categories) by ourselves and the community. We see early access to the dataset as critical in order to understand how others would use the dataset and to derive requirements to develop and refine it.

---

> > ### Comment · Reviewer_jiuK · 2023-08-28
> > **Re: response to reviewer jiuK**
> >
> > Thanks to the author's detailed reply, most of our concerns have been effectively addressed, and I chose to improve my score.

---

> > > ### Author Response · Authors · 2023-08-29
> > >
> > > Thank you for considering our rebuttal text, and we are glad that we were able to provide sufficient clarification on your comments.

---

### Decision · Program_Chairs · 2023-09-22

**Decision:**

Accept (Poster)

**Comment:**

The authors have proposed the DICES dataset as a resrouce toward the evaluation of safety in conversational AI. Specifically, the authors propose two sets of chatbot conversations (DICES-350 and DICES-990), annotated with items pertaining to safety, including the presence of harmful content, unfair bias, misinformation, etc. Crucially, the datasets include fine-grained demographic information about the raters, and the authors include analyses contextualizing the variance, ambiguity, and diversity in safety ratings. The datasets include ratings by both crowd-sourced raters as well as in-house expert annotators (albeit for different proportions in the case of DICES-990).

The reviewers have been favorable in their assessment of the work and the authors have largely incorporated the suggestions of the reviewers to make the work stronger. The authors intend DICES to be a live dataset that will be continually expanded with more granular representation of rater demographics. I agree with their assertion that there is an imminent need for an evaluation dataset for the study of safety with how fast the research around conversation AI and LLMs is moving, and the datasets as presented still constitute a useful resource for the community, so I am recommending to accept this paper. Nevertheless, I have some suggestions for the authors to make the paper stronger in its final version.

**Incorporating a Datasheet for Datasets or equivalent**.  The call for papers (https://neurips.cc/Conferences/2023/CallForDatasetsBenchmarks) explicitly mentions that new datasets must include documentation for intended use, such as a Datasheet for Datasets. Besides being a requirement stipulated by the call, a Datasheet includes items that would address the concerns raised by the ethics reviewer f9LL and reviewer 45Pu.

Specifically, the authors seem resistant to discussing potential misuse of the data: their response to the ethics reveiwer f9LL and reviewer 45Pu on the matter seems to be arguing along the lines of why the pros of sharing this resource with the community outweigh the cons, and how they find it unlikely that someone would *accidentally* use the dataset to create a more biased model. Nevertheless, I still find it prudent to outline how well-intentioned researchers might unintentionally overlook ways in which using the dataset might create more biased models, especially since this is intended to be a live dataset with upcoming updates. This work is interdisciplinary, and potential readers could have varied technical skills. The spirit of the comments by f9LL and 45Pu seems more towards helping readers understand how they can misuse the data rather than suggesting that the resource should not be shared, and I agree. Still, the authors have indeed done this to an extent, especially with the Limitations section, where they discuss the implications of rater demographics, rater disagreements, and the need for further diversity beyond English-speaking locales (also pointed out by reviewer HcmN). While these cover the limitations of the data itself, I would encourage the authors to heed the suggestion of f9LL and 45Pu to expand the discussion with potential pitfalls in *the process of using the data* that can lead to adverse downstream effects (rather than static limitations of the current data) help potential readers with diverse backgrounds. As such, since the call specifies the inclusion of a datasheet as a requirement, I am taking it in good faith that the authors will include such documentation on the intended use and potential misuse.

The authors have stated in the rebuttal (and in the paper) that ` this dataset is not intended for training, but rather for evaluation` (response to jiuK). I suspect the confusion (and subsequent concerns surrouding dataset size) comes from L49-50 under the contributions where the authors mention training (`a more systematic way to account for diverse opinions of safety in model training and evaluation`). It might be simpler to avoid this confusion and explicitly emphasize evaluation again in L49-50.

The references in the checklist are broken, if the authors are going to retain this for the benefit of future readers in the supplement, please fix these. As such, including a Datasheet would address this.